# Distinguishing Among Variants of Primary Progressive Aphasia with a Brief Multimodal Test of Nouns and Verbs

**DOI:** 10.3390/brainsci15101108

**Published:** 2025-10-15

**Authors:** Marco A. Lambert, Melissa D. Stockbridge, Lindsey Kelly, Isidora Diaz-Carr, Voss Neal, Argye E. Hillis

**Affiliations:** 1Department of Neurology, Johns Hopkins University School of Medicine, Baltimore, MD 21287, USA; mlambe25@jh.edu (M.A.L.); mstockb2@jhmi.edu (M.D.S.); lkelly46@jhmi.edu (L.K.); idiazca1@jhu.edu (I.D.-C.); vneal2@jhu.edu (V.N.); 2Department of Physical Medicine and Rehabilitation, Johns Hopkins University School of Medicine, Baltimore, MD 21287, USA; 3Department of Cognitive Science, Krieger School of Arts and Sciences, Johns Hopkins University, Baltimore, MD 21218, USA

**Keywords:** primary progressive aphasia, syntax, naming, diagnosis

## Abstract

Background: Primary Progressive Aphasia (PPA) variants include the non-fluent agrammatic (nfvPPA), logopenic (lvPPA), and semantic (svPPA), which differ in their effects on speech production. However, their impact on modality (oral vs. written) and grammatical word class (nouns vs. verbs) remains controversial. A significant effect of these variables might assist in classification. Materials and Methods: This study used first-visit data from 300 participants with PPA who completed oral and written noun and verb naming (matched in surface word frequency across word class) to test the hypothesis that the three variants show differential impairment on word class or modality. Group differences were evaluated with rank-transformed repeated measures ANOVA. Within individual differences between nouns and verbs and between oral and written modalities were tested with Fisher’s exact tests. Results: A significant modality × variant interaction (*p* = 0.017) was observed. Participants with lvPPA and nfvPPA demonstrated greater oral than written naming, with nfvPPA also performing better on nouns than verbs. Those with svPPA showed no modality or word class effects but had an overall low accuracy. Three participants with svPPA (but no individuals with the other variants) demonstrated significantly (*p* = 0.003) more accurate verb than noun naming. Conclusions: Differing modality and word class patterns characterize PPA variants, with nfvPPA more accurate in nouns than verbs on average. Within individuals, only those with svPPA occasionally showed significantly more proficient verb than noun naming. Grammatical word class effects likely arise at distinct levels of cognitive processing underlying naming.

## 1. Introduction

Primary Progressive Aphasia (PPA) refers to language impairment among people who have a neurodegenerative disorder [1,2]. PPA has been classified into three main variants, which are the non-fluent variant (nfvPPA), the logopenic variant (lvPPA), and the semantic variant (svPPA) [3]. Some patients do not clearly fit the criteria for any one variant and are considered unclassifiable. All these PPA variants share a common symptom of a deficit in word finding and/or production [4]. Language production involves interrelated components, such as lexical retrieval, semantic access, morphosyntactic encoding, and phonological output—each being impacted differently based on the progression and localization of the brain atrophy [5,6]. Understanding the distinct linguistic patterns in each variant is essential not only for diagnostic accuracy, but also for tailoring effective intervention strategies [7]. It helps patients and their families understand their syndrome and what they can expect.

The ability to name nouns and verbs in people with PPA has been a topic of recent interest that elucidates the differences between each variant. Previous research has shown that verb naming is often more impaired than noun naming in nfvPPA and others with posterior frontal damage, while noun naming is more impaired than verb naming in svPPA and others with anterior temporal damage [8]. However, most of the prior research in PPA has been centered on naming pictured objects (nouns) or naming performance tasks that are too broadly categorized, without evaluating the grammatic word class involved, such as word fluency [9]. One older study of 28 people with PPA explored differences between grammatic word class (e.g., nouns vs. verbs) and modalities (oral vs. written) that shape the naming abilities of individuals with PPA [10]. At the time, lvPPA was not recognized as a distinct variant; but the study revealed that on average people with “fluent” PPA had more difficulty with nouns, and people with “nonfluent” PPA had more difficulty with verbs in both modalities. A few individuals in both groups showed modality-specific deficits with either nouns or verbs. The distinctions between these modalities are important, as some individuals with PPA become mute and rely on writing, while others lose the ability to write with their right hand and rely on speech [11]. Others retain the ability to produce isolated words but struggle with their incorporation into grammatically structured speech, depending on the region of the brain that has been damaged and their variant of PPA [12].

Both interactive activation models [13,14] and cognitive neuropsychological models [15] of naming propose distinct levels of processing for the meanings of words (semantics) and for lexical retrieval/output. Within these models, modality effects would occur at the output level. In contrast, grammatical word class effects could arise at the level of semantics (with object vs. actions perhaps relying differentially on certain features such as shape vs. movement) or at the level of output (with grammatical word class represented in distinct neural networks and perhaps different areas of the brain). Individuals with svPPA are thought to have deficits primarily in semantics, while those with lvPPA and nfvPPA are thought to have deficits at the output level. Therefore, we hypothesized that those with svPPA might have word class effects across modalities (but no modality effect), while those with lvPPA or svPPA might have modality effects and perhaps word class effects within modalities.

### 1.1. Verb Naming

A notable feature of nfvPPA patients is that they have affected brain regions dedicated to grammar and sentence formation, particularly in the left posterior frontal lobe [16]. Verb naming deficits associated with left posterior frontal cortex were first observed in people with stroke, particularly those with “agrammatic” or “Broca’s aphasia [17,18,19]. By one account, the resulting agrammatism may be related to difficulties in verb production and inflection, construction of grammatically correct sentences, and properly managing verb tense agreement [20]. Verb inflection errors are particularly common when production of the correct verb depends on its tense form [21,22]. In one study, participants demonstrated more errors when substituting verb inflections given a temporal reference than when given a verb with greater morphological complexity [23]. These findings suggest that the greater challenge for people with nfvPPA may not be the morphological difficulty of producing the verb, but the difficulty in accessing and implementing the syntactic and semantic features required for proper verb production [24].

### 1.2. Noun Naming

Several studies show that noun naming deficits are associated with atrophy in superior portions of the left temporal pole and other anterior temporal regions (critical for semantic processing [25]) in svPPA [26,27] and atrophy in left posterior temporal cortex (critical for lexical output [28] or phonological loop [29]) in lvPPA [30]. These associations between location of neurological damage and grammatical word class mirror those first identified in stroke [17,18,19] and other neurodegenerative diseases [31,32,33].

### 1.3. Verbal and Written Modalities

Prior research has shown a significant effect of modality—written versus oral communication—on naming ability [34]. In nfvPPA, oral naming is sometimes much more severely affected than written naming, as it is affected by the common co-occurrence of progressive apraxia of speech, which results in more effortful and inconsistent speech [35]. Patients with svPPA tend to experience severe impairments in written naming and oral naming, which is likely related to the loss of semantic knowledge affected by the degeneration of the temporal lobe [36]. Although all PPA variants have recognized specific linguistic deficits, they can still experience contrasting performance across modalities [6].

## 2. Materials and Methods

### 2.1. Procedures

This cross-sectional, observational study was a retrospective analysis of prospectively collected data. We reviewed data from a consecutive series of 389 participants diagnosed with PPA at the Johns Hopkins School of Medicine by author AEH between 2008 and June 2025. This population was a convenience sample. Many participants seen in the clinic are only seen for a single diagnostic or confirmatory visit. The diagnosis of PPA and identification of a variant were determined on the basis of a preponderance of the neuropsychological testing, comprehensive neurological evaluation, a review of personal and familial medical history, and consultation with family, if present. Diagnoses were affirmed with magnetic resonance imaging for all patients to determine location and severity of atrophy. Asymmetric frontal atrophy was required for a diagnosis of nfvPPA, primarily left anterior temporal atrophy was required for a diagnosis of svPPA, and left temporo-parietal atrophy was required for a diagnosis of lvPPA. Though 137 patients completed at least one additional visit in which a neuropsychological evaluation occurred (X^−^ = 2.2, σ = 1.6, [1,2,3,4,5,6,7,8,9,10,11] visits), these visits were not included in the present analyses to avoid disproportionate influence of individuals on group statistics. In brief, patients are evaluated in all domains covered in the National Alzheimer’s Coordinating Center (NACC) Uniform Data Set Frontotemporal Lobar Degeneration (FTLD) Module Version 3 module, either with the original tools in the module (some designed by AEH) or with updated, extended, or supplemental tools covering the same domains.

### 2.2. Participants

Individuals were included if they had been classified as having one of the three main PPA variants (based on [3]) and had completed at least one section of the test of oral and written “pure” nouns and “pure” verbs matched for surface word frequency; see below for details. Individuals were excluded from analyses if they had a previous history of stroke, a history of developmental disabilities or fluency disorders, or met criteria for another diagnosis with cognitive implications in addition to PPA. They were also excluded if they had severe apraxia of speech or dysarthria that precluded intelligible oral naming. In addition, patients with unclassifiable PPA or those too severe to have retained the unique characteristics of a specific PPA variant or understand the task were excluded. (Table 1).

See Figure 1 for a flow diagram showing participant screening and inclusion. Percentages refer to the percent of that variant who completed that subtest.

### 2.3. Experimental Test

In the oral and written noun and verb naming subtest used in this study (not part of the NACC FTLD battery), patients are asked to name pictures of common objects and actions and subsequently write down the names of each. Stimuli were taken from a set described by Zingeser and Berndt [17] and included “unambiguous” nouns and verbs. That is, words used as both nouns and verbs (e.g., crack) were excluded. Pictures were black-and-white drawings by a professional artist and were selected for high name agreement (e.g., leaf, melt). Half of the nouns were matched to the “base frequency” of the verbs (frequency of the stem), and half were matched to the “cumulative frequency” of the verbs (frequency of any form of the verbs, such as follow, following, followed). Names were also matched across word class for number of syllables. All stimuli were high imageability; cultural familiarity was not assessed, but the items all had high name agreement and 100% accuracy by healthy controls in the cultural population of the participants of this study (residents of Maryland or surrounding states in the USA). Participants were asked to name the object or action, and the final response was scored. If they produced a correct, but not target word, (e.g., “cook” for bake) they were prompted for another name or a more specific name. Each section has a maximum score of 16 points and is graded based on patient performance as a 1 or 0. Written naming is scored as correct if all letters of the target word were written correctly, and oral naming if all phonemes are spoken in the correct order.

### 2.4. Statistical Analysis

Pursuant to our aim to examine how communication modality influences naming nouns and verbs across the different variants of PPA, we planned for paired analytical approaches. One focused on comparing central tendency across groups, and the other focused on individual likelihood of strong influence of either modality or part of speech on accuracy.

The a priori statistical analysis plan to compare central tendency within whole PPA variant groups was to conduct a repeated measures analysis of variance. However, it was noted during the testing of assumptions that the null hypothesis that the observed covariance matrices of the dependent variables was equal across groups was not supported (Box’s M = 117.5, F(20, 84,880) = 5.7, *p* < 0.001), nor was the error covariance matrix of the orthonormalized dependent variable proportional to an identity matrix (Mauchly’s W = 1). These violations of normality were due to the considerable frequency of ceiling effects on the task. The task within the FTLD module intentionally uses vocabulary that has high frequency within the English language, low age of acquisition, and moderate to low semantic and phonological complexity, leading most healthy respondents to perform at or near ceiling and those with diagnoses, often in early disease, to demonstrate clear negative skewness. In response to this violation of assumptions, task performance data were adjusted rank-transformed prior to analysis using the Excel macro ART-ANOVA, rendering the analysis non-parametric and robust to the violation [37] (http://derwinchan.iwopop.com/Art-ANOVA, accessed on 16 July 2025, Chan, D. K. C. (2013). Art-ANOVA [Computer software]. Available from www.derwinchan.com, accessed on 16 July 2025). Once the data were transformed, age was entered as a covariate in recognition of the significant differences in age by group anticipated in light of prior work and observed in the sample. Where significant effects were observed, pairwise contrasts were examined using Bonferroni post hoc tests.

To examine the individual likelihood of differences due to modality or part of speech, we conducted Fisher’s exact tests. First, all four pairwise differences (oral versus written nouns, oral versus written verbs, oral nouns versus verbs, and written nouns versus verbs) were calculated. Due to a concern for inflated Type I error, rather than testing all four contrasts, mean differences were calculated by modality irrespective of part of speech and by part of speech irrespective of modality. These were used for hypothesis testing, and constituent contrasts were considered only if the main effect was significant. Bonferroni-corrected post hoc tests of residuals were used to provide interpretation of the results of the resulting 3 × 2 and 3 × 3 χ^2^ tests.

Some participants had missing data, especially for written names, due to motor difficulties in writing or inconvenience in assessing writing due to on-line testing. We evaluated differences between those with and without missing data for any of the experimental subtests, using *t*-tests and chi-squared tests. For all tests, we used an alpha level of 0.05 and corrected for multiple comparisons with Bonferroni correction.

## 3. Results

Participant characteristics are reported in Table 2. Groups differed in age; F(2, 297) = 5.1, *p* = 0.007. This difference was driven by patients with lvPPA being older on average than those with svPPA (Mean Difference = 3.7, Standard Error = 1.2, *p* = 0.005; 95% Confidence Interval [0.9–6.5]). This age difference is common across PPA variants. Other characteristics were similar across groups.

### 3.1. Analysis of Performance by Variant

Mean performance across conditions is summarized in Figure 2. The repeated measures analysis of variance on transformed data identified a significant interaction between modality (oral or written) and PPA variant, Pillai’s Trace = 0.036, F(2, 222) = 4.1, *p* = 0.017, η^2^_P_ = 0.036 when controlling for age. The interaction between part of speech (noun or verb) and PPA variant approached significance; Pillai’s Trace = 0.026, F(2, 222) = 2.9, *p* = 0.057, η^2^_P_ = 0.026. There was no significant main effect of modality or part of speech, and no other significant interaction effects. The between-subjects main effects of age and variant were not significant.

The significant interaction between modality and PPA variant was investigated using Bonferroni post hoc tests within each variant considered independently. Among those with lvPPA, there was a significant main effect of modality, F(2, 117) = 24.2, *p* < 0.001, η^2^_P_ = 0.17. The main effect of modality was the result of oral naming being more accurate than written naming. The main effect of part of speech and the interaction between part of speech and modality were not significant. Among those with nfvPPA, there was a significant main effect of modality F(1, 49) = 5.5, *p* = 0.023, η^2^_P_ = 0.10 and part of speech, F(1, 49) = 4.4, *p* = 0.042, η^2^_P_ = 0.08, with no significant interaction. Oral naming was more accurate than written naming, and noun naming was more accurate than verb naming. Among those with svPPA, neither modality nor part of speech nor the interaction significantly impacted naming accuracy. As anticipated, accuracy overall on naming was depressed relative to the other variants.

### 3.2. Analysis of Performance Within Individuals

Participants were identified as demonstrating a significant difference in performance based on modality, based on part of speech, both, or neither. No patient demonstrated significantly better written performance than oral performance, although this result may reflect the fact that patients with severe apraxia of speech or dysarthria precluding intelligible responses were excluded (all of whom had nfvPPA). Only one patient (with lvPPA) showed a significantly (by Fisher’s exact test) more accurate performance in oral than written naming (87.5% vs. 25% accurate; *p* < 0.0001).

The likelihood of differences between nouns and verbs was significantly different depending on participants’ PPA variant, χ^2^(4) = 9.6, exact *p* = 0.048. Three participants were significantly more accurate in naming verbs than nouns, all of whom had the semantic variant. The greater accuracy for verbs than nouns was significant by 2-sided Fisher’s exact test in all three participants (87.5% vs. 46.9%; *p* = 0.001; 56.2% vs. 6.3%; *p* = 0.006; 37.5% vs. 0%; *p* = 0.018). Bonferroni-corrected post hoc tests of adjusted residuals demonstrated that this was significantly more than expected by chance (*p* = 0.003). In contrast, similar proportions of participants across variants demonstrated higher accuracy in naming nouns (10–14%, n.s.).

## 4. Discussion

In this study, we examined the different effects that PPA variants can have on ability to orally name and write the names of objects and actions, to test the hypothesis that those with svPPA might have word class effects across modalities (but no modality effect), while those with lvPPA or svPPA might have modality effects and perhaps word class effects within modalities. This hypothesis was largely supported by the results, although we identified quite heterogeneous individual patterns of performance within variants.

Our analyses of group tendency in a large sample of participants provided confirmation of modality effects primarily in nfvPPA and lvPPA but not in svPPA. There was a significant interaction between response modality and PPA variant. This appeared driven by the contrast between participants with lvPPA and nfvPPA, whose average performance in oral naming was greater than written naming and contrasted with participants with svPPA, for whom modality did not significantly impact performance. Rather, those with svPPA showed the most depressed naming overall. Previous studies have shown that many people with nfvPPA show better performance with written than oral naming [10], while our study showed (on average) the opposite. Both findings support modality effects in nfvPPA. However, in this study, but not in our previous study, we excluded participants with severe motor speech deficits (dysarthria or apraxia of speech precluding intelligible responses), which may account for the difference in findings. Furthermore, while some participants showed significant differences between noun and verb naming, most individuals did not. In prior studies, a higher percentage of participants showed significant differences between word classes [10]. This discrepancy likely reflects a shift in criteria for PPA. Originally, the diagnosis of PPA required progressive impairment in language for at least two years before impairment in other domains of cognition (other than apraxia) [38]. Now, the diagnosis of PPA requires a predominance of deficits in language at onset and in the initial stage of the condition [3]. It is likely that early studies of PPA included individuals with much more focal atrophy (which can affect nouns more than verbs or vice versa, or oral more than written naming or vice versa) [35], while current studies include those with more diffuse atrophy that affects both word classes and modalities.

Individuals with Alzheimer’s disease have significantly greater deficits in written than in oral naming [39,40], but no effect of part of speech [41]. This holds true among those with the lvPPA syndrome, which is typically due to Alzheimer’s disease [3]. Prior studies that have investigated the modality in nfvPPA show that part of speech is affected, with greater difficulty producing verbs than nouns in both modalities, and an even greater difficulty when these verbs relied on tense processing [10,23,32,36]. Participants with nfvPPA described in the present study had more difficulty naming verbs than nouns on average, as has been reported in prior literature on agrammatism [10,31,32]. Severe word-finding difficulties are a hallmark of svPPA, irrespective of part of speech or modality, due to a multiplicity of factors that affect the semantic system [27].

Considering only those differences within participants that were significant by Fisher’s exact test highlighted that stronger verb than noun naming occurred only in svPPA, and not in the other variants. The improbable significant difference only seen in svPPA results in a high likelihood that these participants can be confidently classified using the oral and written noun and verb naming test, as long as word comprehension deficits are also present. As there was no modality effect in svPPA, including those with significantly better verbs than nouns, the word class effect most likely arises at the level of semantics (shared by oral and written naming). Various proposals have been put forward to explain word class effects in semantic processing, including distinct regions of processing the meaning of objects (anterior temporal) and the meaning of actions (inferior and middle frontal, parietal, and posterior temporal) [31,42]. However, see [43] for a refutation of this proposal. Alternative accounts include difference between nouns and verbs in animacy, sensory versus somatomotor features, and so on [42]. Our data do not help adjudicate between these proposals. Word class effects can also arise due to stimulus features, but these generally favor nouns (objects) over verbs (actions), as actions are more difficult to portray unambiguously in pictures. Word class effects can also occur at the level of lexical selection [44] (which is likely the case for modality-specific word class effects, which have been reported in nfvPPA) [35].

Only one individual (with lvPPA) showed a significant modality effect, favoring oral over written naming. Individuals with all forms of PPA frequently have spelling impairment, sometimes as the earliest deficit. Spelling deficits across variants occur during output processes—at the level of orthographic lexical selection and/or phonology-to-orthography conversion, or orthographic buffer [45].

### 4.1. Limitations

We were unable to fully evaluate our hypothesis because fewer people completed the writing subtests than the oral subtests. There was no single contributor to this pattern of missing data, but evaluation time constraints, fatigue, pain when writing due to arthritis, and other non-cognitive conditions likely accounted for some of the missingness. Thus, although average performance across groups supported an oral performance bias, the clinical interpretability of that finding is limited by inconsistency with individual data and missing data. Based on our findings, it is clear that the part of speech effect can be observed when testing participants in the oral modality. However, we recommend that clinicians interested in fully characterizing the strengths and weaknesses of their participants under investigation for PPA do include both modalities of response.

Another limitation is that the test used stimuli with relatively high word frequency, which may have resulted in ceiling effects that underestimated the frequency of dissociations between nouns and verbs. Future studies may want to include nouns and verbs with a great range in frequency of occurrence. Furthermore, the study included only English-speaking participants from a single clinic; the absence of multilingual or cross-site validation limits the generalizability of the findings.

### 4.2. Conclusions

The findings from the present investigation indicate the potential usefulness of evaluating oral and written nouns and verbs in the diagnosis of PPA and its variants in contexts of care where neuroimaging and plasma biomarkers are not available to achieve a gold-standard diagnosis. More importantly, given the individuality of neurodegeneration that leads to the development of PPA and the limited resources in many diverse contexts, it is imperative to develop diagnostic criteria that are both reliable and accessible, to aid in prognostication for family members. However, given the heterogeneity of our results within variants, the finding of a modality effect or word class effect cannot be used alone for classification. Significantly more accurate naming of verbs than nouns (in confrontation naming) and lack of a modality effect will favor the likelihood of svPPA (like more accurate reading or spelling of regular than irregular words) but would only strengthen the classification in the presence of impaired word comprehension. A significant modality (oral versus written) effect would favor lvPPA or nfvPPA in the presence of other key features of these variants. Irrespective of the variant, differential impairment of nouns or verbs within an individual is also important as it may also guide treatment of naming deficits [46]. Several treatments have been designed to target verbs (e.g., VNeST ) [47,48] or to improve nouns and verbs separately [49]. Likewise, treatments to specifically target spelling or written naming (as well as the many that target oral naming) can be useful in PPA [50,51]. Hence, clinicians are encouraged to evaluate oral and written nouns and verbs in people with PPA (our short multimodality test, developed for English speakers, is available at (https://score.jhmi.edu/, accessed on 14 October 2025).

## Figures and Tables

**Figure 1 brainsci-15-01108-f001:**
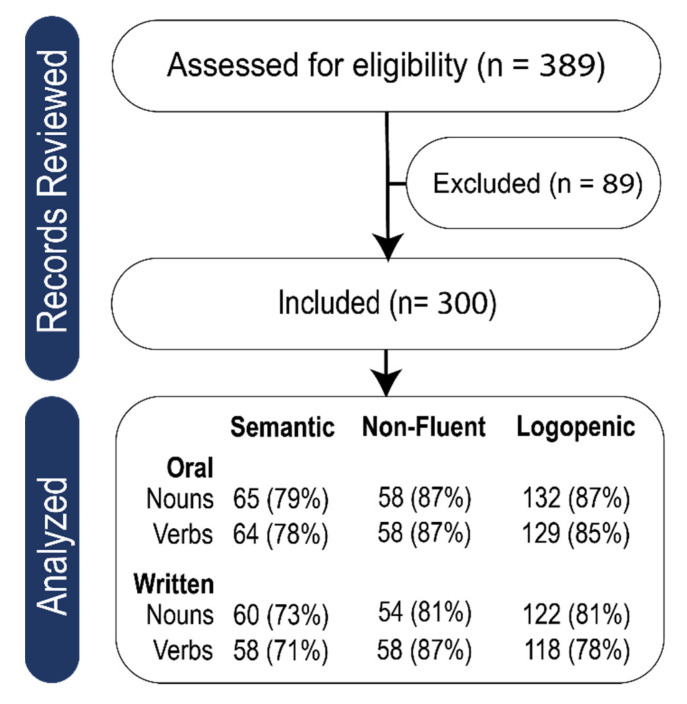
Flow diagram showing participant screening and inclusion.

**Figure 2 brainsci-15-01108-f002:**
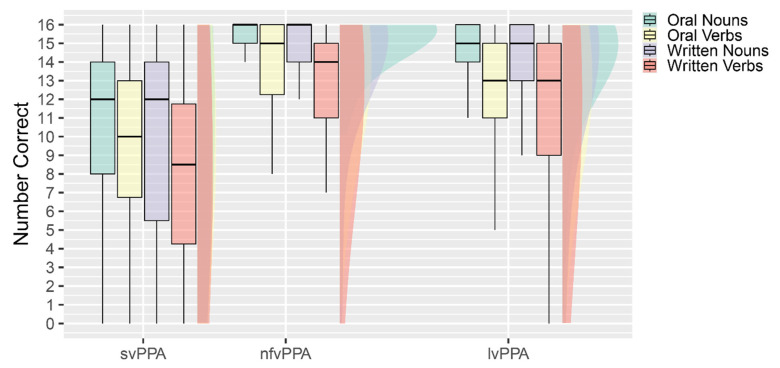
Mean correct names in each subtest by each variant.

**Table 1 brainsci-15-01108-t001:** Records reviewed and application of exclusionary criteria.

Criterion	Count
Previous history of stroke	16
Speech impediment, apraxia of speech, or dysarthria that interfered with intelligibility	6
Previous history of learning disabilities	2
Diagnosis did not include PPA	18
PPA too progressed for task	2
PPA identified but deemed unclassifiable into three classic variants	44
Evaluation data were missing	1
**Records excluded**	89

**Table 2 brainsci-15-01108-t002:** Participant characteristics.

	Semantic	Non-Fluent	Logopenic
Count	82	67	151
Age *	67 (8)	70 (9)	71 (8)
Sex (Female:Male)	51:31	30:37	89:62
Education (*N* = 141)	16 (2)	16 (3)	16 (3)
Handedness (Right:Left; *N* = 168)	42:4	41:6	70:5

Mean (Standard Deviation) unless otherwise noted. * *p* < 0.05. Education and handedness were recorded for many, but not all, patients as part of their cognitive neurological evaluation. There were no differences between participants with missing data (*n* = 62) on one or more subtests and those without missing data (*n* = 238), in terms of age (t = −0.11; df298; *p* = 0.92), education (t = −0.94; df231; *p* = 0.35), handedness (χ^2^ = 0.95; df1; *p* = 0.33), or distribution of the 3 variants (χ^2^ = 2.0; df1; *p* = 0.37).

## Data Availability

The data presented in this study are available on request from the corresponding author due to restrictions imposed by the Data Trust committee of Johns Hopkins University School of Medicine.

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
