# Peer review of "Distinguishing Among Variants of Primary Progressive Aphasia with a Brief Multimodal Test of Nouns and Verbs"

_brainsci, 2025, doi:10.3390/brainsci15101108_

Round 1

Reviewer 1 Report

Comments and Suggestions for Authors

First of all, thank you very much for the opportunity to review this manuscript.

The topic of this study is relevant due to the simplicity and specificity of the proposed diagnostic criteria in the differential diagnosis of various forms of primary progressive aphasia.

The title matches the content.

It is necessary to reformulate the abstract, indicating the background, materials and methods, results and conclusions.

The introduction is written in a very interesting and informative manner and touches upon the main distinguishing features of the various forms of PPA. However, I think it would be very informative if you could add information about similar works from literature and what makes your work unique.

In my opinion, the purpose of this study could be improved if some clarifications were made. “ “is to better understand how communication modality influences”

Please change the section title from "Methods" to "Materials and Methods".

A methods section should be provided point by point with more detailed information.

Please indicate the design of your study.

Name the method for determining the sample size with the definition of its alpha and power.

In line “see [13] for details”.  If necessary, please include these details in your article. Moreover, the article [13] is not publicly available.

Add please, study population selection diagram.

The statistical analysis section is very well written with detailed explanations.

The authors presented their results using, in my opinion, very precise and effective statistical tools. However, diagrams are missing for this section.

In discussion: “In this study, we examined the different effects that PPA variants can have on language production and proficiency. What was of particular interest was how people with different variants of PPA would be affected in their ability to orally name and write the names of objects and actions, as this was potentially informative to the differential diagnostic process. Participants were shown two sets of images depicting sixteen nouns and verbs respectively and were asked to provide an oral and written name for each image. “In my opinion, this paragraph refers to materials and methods.

In discussion: “However, in this study, we excluded participants with severe motor speech deficits (dysarthria or apraxia of speech precluding intelligible responses)” Why is this information not included in Table 1?

In discussion: Add please references after” Previous studies have shown that many people with nfvPPA show better performance with oral than written naming”

In discussion: Add please references after “In prior studies, a higher percentage of participants showed significant differences (Hillis, 2004)”

In discussion: Add please references after “Originally, the diagnosis of PPA required progressive impairment in language for at least two years before impairment in other domains of cognition (other than apraxia).

In discussion: Add please references after  “Now, the diagnosis of PPA requires a predominance of deficits in language over time”

In discussion: Add please references after  “It is likely that early studies of PPA included individuals with much more focal atrophy (which can affect nouns more than verbs, or oral versus written naming)”

The conclusions should specifically highlight the distinguishing characteristics between the three APF groups using a short multimodal test of nouns and verbs.

Author Response

Comment 1. It is necessary to reformulate the abstract, indicating the background, materials and methods, results and conclusions.

Response 1. We restructured the abstract.

Comment 2. The introduction is written in a very interesting and informative manner and touches upon the main distinguishing features of the various forms of PPA. However, I think it would be very informative if you could add information about similar works from literature and what makes your work unique.

Response 2. We have added many more references to similar works, and how our work is unique, in the introduction and discussion.

Comment 3. In my opinion, the purpose of this study could be improved if some clarifications were made. “ “is to better understand how communication modality influences”

Response 3. We clarified that our purpose was to evaluate a specific hypothesis: “Therefore, we hypothesized that those with svPPA might have word class effects across modalities (but no modality effect), while those with lvPPA or svPPA might have modality effects and perhaps word class effects within modalities.”

Comment 4. Please change the section title from "Methods" to "Materials and Methods".

Response 4. Done

Comment 5. A methods section should be provided point by point with more detailed information.

Response 5. We added point by point procedures and more detailed information.

Comment 6. Please indicate the design of your study.

Response 6. Added

Comment 7. Name the method for determining the sample size with the definition of its alpha and power.

Response 7. We added that the sample was a convenience sample (all available cases that met criteria) and added the alpha level.

Comment 8. In line “see [13] for details”.  If necessary, please include these details in your article. Moreover, the article [13] is not publicly available.

Response 8. We added details regarding the test:

In the Oral and Written Noun and Verb Naming Subtest used in this study (not part of the NACC FTLD battery), patients are asked to name pictures of common objects and actions and subsequently write down the names of each. Stimuli were taken from a set described by Zingeser and Berndt (reference added), and included “unambiguous” nouns and verbs.  That is, words used as both nouns and verbs (e.g. crack) were excluded.  Pictures were black and white drawings by a professional artist, and were selected for high name agreement (e.g. leaf, melt). Half of the nouns were matched to the “base frequency” of the verbs (frequency of the stem), and half were matched to the “cumulative frequency” of the verbs (frequency of any form of the verbs, such as pour, pouring, poured ).  Names were also matched across word class for number of syllables. All stimuli were high imageability; cultural familiarity was not assessed, but the items all had high name agreement and 100% accuracy by healthy controls in the cultural population of the participants of this study (residents of Maryland, USA). For stimuli, see https://score.jhmi.edu/downloads. Participants were asked to name the object or action, and the final response was scored.  If they produced a correct, but not target word, (e.g. “cook” for bake) they were prompted for another name or a more specific name.  Each section has a maximum score of 16 points and is graded based on patient performance as a 1 or 0. Written naming is scored as correct if all letters of the target word were written correctly, and Oral naming if all phonemes are spoken in the correct order. 

Comment 9. The statistical analysis section is very well written with detailed explanations.

Response 9. Thank you.

Comment 10. The authors presented their results using, in my opinion, very precise and effective statistical tools. However, diagrams are missing for this section.

Response 10. We added the diagram.

Comment 11. In discussion: “In this study, we examined the different effects that PPA variants can have on language production and proficiency. What was of particular interest was how people with different variants of PPA would be affected in their ability to orally name and write the names of objects and actions, as this was potentially informative to the differential diagnostic process. Participants were shown two sets of images depicting sixteen nouns and verbs respectively and were asked to provide an oral and written name for each image. “In my opinion, this paragraph refers to materials and methods.

Response 11. We agree, and we deleted it from the Discussion

Comment 12. In discussion: “However, in this study, we excluded participants with severe motor speech deficits (dysarthria or apraxia of speech precluding intelligible responses)” Why is this information not included in Table 1?

Response 12. It was included as the second line, but we clarified by removing the words “previous history of” and added “apraxia of speech” as a specific type speech impediment. It now reads,

“Speech impediment, apraxia of speech, or dysarthria that interfered with intelligibility”

Comment 13.In discussion: Add please references after” Previous studies have shown that many people with nfvPPA show better performance with oral than written naming”

Response 13. References were added.

Comment 14. In discussion: Add please references after “In prior studies, a higher percentage of participants showed significant differences (Hillis, 2004)”

Response 14. References were added and (Hillis, 2004) was deleted.

Comment 15. In discussion: Add please references after “Originally, the diagnosis of PPA required progressive impairment in language for at least two years before impairment in other domains of cognition (other than apraxia).

Response 15. We added the appropriate reference.

Comment 16. In discussion: Add please references after  “Now, the diagnosis of PPA requires a predominance of deficits in language over time ”

Response 16. We added the appropriate reference.

Comment 17. In discussion: Add please references after  “It is likely that early studies of PPA included individuals with much more focal atrophy (which can affect nouns more than verbs, or oral versus written naming )”

Response 17. We added the appropriate reference.

Comment 18. The conclusions should specifically highlight the distinguishing characteristics between the three APF groups using a short multimodal test of nouns and verbs.

Response 18. We revised the conclusions as suggested:

The findings from the present investigation indicate the potential usefulness of evaluating oral and written nouns and verbs in the diagnosis of PPA and its variants in contexts of care where neuroimaging and plasma biomarkers are not available to achieve a gold-standard diagnosis. More importantly, given the individuality of neurodegeneration that leads to the development of PPA and the limited resources in many diverse contexts, it is imperative to develop diagnostic criteria that are both reliable and accessible, to aid in prognostication for family members. However, given the heterogeneity of our results within variants, the finding of a modality effect or word class effect cannot be used alone for classification. Significantly more accurate naming of verbs than nouns (in confrontation naming) and lack of a modality effect will favor the likelihood of svPPA (like more accurate reading or spelling of regular than irregular words) but would only strengthen the classification in the presence of impaired word comprehension. A significant modality (oral versus written) effect would favor lvPPA or nfvPPA in the presence of other key features of these variants. Irrespective of the variant, differential impairment of nouns or verbs within an individual is also important as it may also guide treatment of naming deficits [45]. Several treatments have been designed to target verbs (e.g. VNeST ) [46,47] or to improve nouns and verbs separately [48].  Likewise, treatments to specifically target spelling or written naming (as well as the many that target oral naming) can be useful in PPA [49,50].  Hence, clinicians are encouraged to evaluate oral and written nouns and verbs in people with PPA (our short multimodality test, developed for English speakers, is available at https://score.jhmi.edu/downloads).

Reviewer 2 Report

Comments and Suggestions for Authors
  1. The claim that verb > noun naming in svPPA could serve as a diagnostic marker is based on only three participants; this is statistically fragile and the diagnostic implication is overstated.
  2. The dataset, while large, comes from a single institution with English-speaking participants; the lack of multilingual or cross-site validation limits the generalisability of the findings.
  3. The paper notes that many participants did not complete the written modality due to fatigue or arthritis, but it is unclear how these missing data were handled in the analyses; listwise exclusion could bias results toward less impaired patients.
  4. The use of ART-ANOVA is appropriate for ceiling effects, but effect sizes are small (e.g., η²p = .036 for modality × variant interaction) and their clinical significance is not discussed; confidence intervals should be provided consistently.
  5. The threshold of a five-point difference for defining significant within-individual contrasts is arbitrary; a clear justification or reference is needed.
  6. The introduction spends considerable space on background already well-established (variant criteria, atrophy patterns); it could be streamlined to better focus on the novel aspects of the study.
  7. The Oral and Written Noun and Verb Naming Subtest is described only briefly; more detail on stimulus validation (frequency, imageability, cultural familiarity) is necessary to evaluate the robustness of the task.
  8. The discussion of why some svPPA participants performed better on verbs than nouns is speculative and lacks engagement with alternative explanations such as lexical selection, semantic category familiarity, or stimulus effects.
  9. The presentation would benefit from additional visualisation of the distribution of scores (e.g., violin or density plots) rather than relying mainly on group means.
  10. The paper concludes with strong clinical implications, yet given the small effect sizes and limited generalisability, the claims should be moderated to reflect that the findings are suggestive rather than definitive.

Author Response

Comment 1. The claim that verb > noun naming in svPPA could serve as a diagnostic marker is based on only three participants; this is statistically fragile and the diagnostic implication is overstated.

Response 1. We have revised the discussion and conclusion, such that the diagnostic implication is no longer overstated.

Comment 2. The dataset, while large, comes from a single institution with English-speaking participants; the lack of multilingual or cross-site validation limits the generalisability of the findings.

Response 2. We add this as a limitation of the study.

Comment 3. The paper notes that many participants did not complete the written modality due to fatigue or arthritis, but it is unclear how these missing data were handled in the analyses; listwise exclusion could bias results toward less impaired patients.

The use of ART-ANOVA is appropriate for ceiling effects, but effect sizes are small (e.g., η²p = .036 for modality × variant interaction) and their clinical significance is not discussed; confidence intervals should be provided consistently.

Response 3. We agree, and now discuss the clinical significance

Comment 4. The threshold of a five-point difference for defining significant within-individual contrasts is arbitrary; a clear justification or reference is needed.

Response 4. We actually just used the 5-point difference to screen for individuals with potentially significant effects.  We actually used a statistically significant difference by Fisher’s exact to identify individuals with significant effects. We clarify this in the methods and results.

Comment 5. The introduction spends considerable space on background already well-established (variant criteria, atrophy patterns); it could be streamlined to better focus on the novel aspects of the study.

Response 5. We deleted much of the well-established background .

Comment 6. The Oral and Written Noun and Verb Naming Subtest is described only briefly; more detail on stimulus validation (frequency, imageability, cultural familiarity) is necessary to evaluate the robustness of the task.

Response 6. We now describe the test in detail. We added:

 In the Oral and Written Noun and Verb Naming Subtest used in this study (not part of the NACC FTLD battery), patients are asked to name pictures of common objects and actions and subsequently write down the names of each. Stimuli were taken from a set described by Zingeser and Berndt (reference added), and included “unambiguous” nouns and verbs.  That is, words used as both nouns and verbs (e.g. crack) were excluded.  Pictures were black and white drawings by a professional artist, and were selected for high name agreement (e.g. leaf, melt). Half of the nouns were matched to the “base frequency” of the verbs (frequency of the stem), and half were matched to the “cumulative frequency” of the verbs (frequency of any form of the verbs, such as pour, pouring, poured ).  Names were also matched across word class for number of syllables. All stimuli were high imageability; cultural familiarity was not assessed, but the items all had high name agreement and 100% accuracy by healthy controls in the cultural population of the participants of this study (residents of Maryland, USA). For stimuli, see https://score.jhmi.edu/downloads. Participants were asked to name the object or action, and the final response was scored.  If they produced a correct, but not target word, (e.g. “cook” for bake) they were prompted for another name or a more specific name.  Each section has a maximum score of 16 points and is graded based on patient performance as a 1 or 0. Written naming is scored as correct if all letters of the target word were written correctly, and Oral naming if all phenomes are spoken in the correct order. 

Comment 7. The discussion of why some svPPA participants performed better on verbs than nouns is speculative and lacks engagement with alternative explanations such as lexical selection, semantic category familiarity, or stimulus effects.

Response 7. We add: As there was no modality effect in svPPA, including those with significantly better verbs than nouns, the word class effect most likely arises at the level of semantics (shared by oral and written naming).  Various proposals have been put forward to explain word class effects in semantic processing, including distinct regions of processing the meaning of objects (anterior temporal) and the meaning of objects (inferior and middle frontal, parietal, and posterior temporal ). (references added).  However, see Vigiocco et al . for a refutation of this proposal. Alternative accounts include difference between nouns and verbs in animacy, sensory versus somatomotor features, and so on (reference added). Our data do not help adjudicate between these proposals.  Word class effects can also arise due to stimulus features, but these generally favor nouns (objects) over verbs (actions), as actions are more difficult to portray unambiguously in pictures.  Word class effects can also occur at the level of lexical selection (reference added)  (which is likely the case for modality-specific word class effects, which have been reported in nfvPPA ).(reference added)

Comment 8. The presentation would benefit from additional visualisation of the distribution of scores (e.g., violin or density plots) rather than relying mainly on group means.

Response 8. We added a graph with violin plots.

Comment 9. The paper concludes with strong clinical implications, yet given the small effect sizes and limited generalisability, the claims should be moderated to reflect that the findings are suggestive rather than definitive.

Response 9.  We modified the conclusion as follows: (with appropriate references in the text)

The findings from the present investigation indicate the potential usefulness of evaluating oral and written nouns and verbs in the diagnosis of PPA and its variants in contexts of care where neuroimaging and plasma biomarkers are not available to achieve a gold-standard diagnosis. More importantly, given the individuality of neurodegeneration that leads to the development of PPA and the limited resources in many diverse contexts, it is imperative to develop diagnostic criteria that are both reliable and accessible, to aid in prognostication for family members. However, given the heterogeneity of our results within variants, the finding of a modality effect or word class effect cannot be used alone for classification. Significantly more accurate naming of verbs than nouns (in confrontation naming) and lack of a modality effect will favor the likelihood of svPPA (like more accurate reading or spelling of regular than irregular words), but would only strengthen the classification in the presence of impaired word comprehension. A significant modality (oral versus written) effect would favor lvPPA or nfvPPA in the presence of other key features of these variants. Irrespective of the variant, differential impairment of nouns or verbs within an individual is also important as it may also guide treatment of naming deficits . Several treatments have been designed to target verbs (e.g. VNeST ) or to improve nouns and verbs separately .  Likewise, treatments to specifically target spelling or written naming (as well as the many that target oral naming) can be useful in PPA .  Hence, clinicians are encouraged to evaluate oral and written nouns and verbs in people with PPA (our short multimodality test, developed for English speakers, is available at https://score.jhmi.edu/downloads).

Reviewer 3 Report

Comments and Suggestions for Authors

The manuscript makes a valuable contribution by examining how different variants of PPA affect noun and verb naming across oral and written modalities. One of the key strengths of the study lies in the large sample size of participants, which is not necessarily common in PPA research and adds weight to the findings. The inclusion of both group-level and individual-level analyses is another strength, as it highlights both general trends and meaningful variability across participants. The methodological transparency is commendable, particularly the decision to adjust for violations of statistical assumptions by using rank-transformed ANOVA, which is appropriate for the skewed data. The findings, especially the identification of a small subset of semantic variant PPA patients with better verb than noun naming, are clinically relevant and potentially useful for differential diagnosis when more advanced neuroimaging or biomarkers are not available.

Several aspects of the manuscript could be improved to enhance its clarity, rigour, and broader impact. These are for the authors’ reference only. First, the theoretical framing could be deepened, as the introduction largely reiterates prior findings on noun–verb and modality differences without clearly articulating how this study advances or challenges existing models of lexical access and grammatical processing in PPA. The discussion would benefit from a more explicit connection between the observed patterns and current neurocognitive theories of language production. For example, situating the findings in relation to interactive activation or dual-route models could help clarify their theoretical significance.

The methodology section could also provide more detail about the reliability and scoring of the oral and written naming tasks. Although the task is described as part of a well-established battery, an additional explanation of how inter-rater reliability was ensured would reassure readers about the robustness of the results. Similarly, while the use of high-frequency words makes the test clinically feasible, the resulting ceiling effects may have masked subtler differences between groups. The implications of this limitation should be expanded upon, with suggestions for how future research might incorporate more graded stimuli to capture a fuller range of performance.

The paper clearly has both theoretical and practical contributions. In terms of practical contributions, the paper could go further in explaining how the findings might inform clinical assessment and intervention. For instance, the identification of verb–noun dissociations in a subset of svPPA patients could be translated into specific recommendations for diagnostic protocols, particularly for populations of diverse backgrounds, e.g., CALD. A more detailed discussion of how clinicians can use the modality and word-class profiles in real-world settings would help the study be more actionable.

I think, overall, this is a strong and clinically relevant study

Author Response

Comment 1. Several aspects of the manuscript could be improved to enhance its clarity, rigour, and broader impact. These are for the authors’ reference only. First, the theoretical framing could be deepened, as the introduction largely reiterates prior findings on noun–verb and modality differences without clearly articulating how this study advances or challenges existing models of lexical access and grammatical processing in PPA. The discussion would benefit from a more explicit connection between the observed patterns and current neurocognitive theories of language production. For example, situating the findings in relation to interactive activation or dual-route models could help clarify their theoretical significance.

Response 1. We agree, and added a theoretical framework in the introduction and discussion, referring to interactive activation and cognitive neuropsychological models of naming.

Comment 2. The methodology section could also provide more detail about the reliability and scoring of the oral and written naming tasks. Although the task is described as part of a well-established battery, an additional explanation of how inter-rater reliability was ensured would reassure readers about the robustness of the results. Similarly, while the use of high-frequency words makes the test clinically feasible, the resulting ceiling effects may have masked subtler differences between groups. The implications of this limitation should be expanded upon, with suggestions for how future research might incorporate more graded stimuli to capture a fuller range of performance.

Response 2. Because only 1 final response was considered correct for each stimulus, inter-rater reliability in scoring 20 randomly selected tests was 100% point-to-point agreement.  We added the limitation of word frequency, under limitations, as follows:  “The test used stimuli with relatively high word frequency, which may have resulted in ceiling effects that underestimated the frequency of dissociations between nouns and verbs.  Future studies may want to include nouns and verbs with a great range in frequency of occurrence.”

Comment 3. The paper clearly has both theoretical and practical contributions. In terms of practical contributions, the paper could go further in explaining how the findings might inform clinical assessment and intervention. For instance, the identification of verb–noun dissociations in a subset of svPPA patients could be translated into specific recommendations for diagnostic protocols, particularly for populations of diverse backgrounds, e.g., CALD. A more detailed discussion of how clinicians can use the modality and word-class profiles in real-world settings would help the study be more actionable.I think, overall, this is a strong and clinically relevant study.

Response 3. We agree, and added in the conclusion how the results can be more actionable, as follows:

The findings from the present investigation indicate the potential usefulness of evaluating oral and written nouns and verbs in the diagnosis of PPA and its variants in contexts of care where neuroimaging and plasma biomarkers are not available to achieve a gold-standard diagnosis. More importantly, given the individuality of neurodegeneration that leads to the development of PPA and the limited resources in many diverse contexts, it is imperative to develop diagnostic criteria that are both reliable and accessible, to aid in prognostication for family members. However, given the heterogeneity of our results within variants, the finding of a modality effect or word class effect cannot be used alone for classification. Significantly more accurate naming of verbs than nouns (in confrontation naming) and lack of a modality effect will favor the likelihood of svPPA (like more accurate reading or spelling of regular than irregular words) but would only strengthen the classification in the presence of impaired word comprehension. A significant modality (oral versus written) effect would favor lvPPA or nfvPPA in the presence of other key features of these variants. Irrespective of the variant, differential impairment of nouns or verbs within an individual is also important as it may also guide treatment of naming deficits [45]. Several treatments have been designed to target verbs (e.g. VNeST ) [46,47] or to improve nouns and verbs separately [48].  Likewise, treatments to specifically target spelling or written naming (as well as the many that target oral naming) can be useful in PPA [49,50].  Hence, clinicians are encouraged to evaluate oral and written nouns and verbs in people with PPA (our short multimodality test, developed for English speakers, is available at https://score.jhmi.edu/downloads).

Round 2

Reviewer 1 Report

Comments and Suggestions for Authors

Thank you very much for the opportunity to review this manuscript again.

The changes in the title are much more in line with the content.

Added information and more up-to-date links in introduction have greatly increased the importance and relevance of this section.

In my opinion, reformulating the purpose made it more consistent and specific.

Changes in materials and methods, particularly with regard to sample size determination and item-by-item description of research methods, increased the reliability of the results obtained and provided greater clarity in the methods used.

The added figure 2, in my opinion, made the results more clear and understandable.

The revised version's conclusions are more specific and better aligned with the findings, with an emphasis on addressing remaining issues within the study's purpose.

Author Response

No further comments have been made.

Reviewer 2 Report

Comments and Suggestions for Authors

Thank you for addressing several of the previous comments, particularly those related to sample description, missing-data handling, and alternative interpretations in the discussion. However, the following points remain unaddressed and require further revision: (1) the justification for the within-individual contrast threshold (previously noted as arbitrary) is still missing; (2) the introduction continues to devote substantial space to well-established background rather than focusing on the novel contribution of the study; and (3) the presentation of results still lacks richer visualisation of score distributions (e.g., violin or density plots) beyond group means. Clarifying these aspects would considerably strengthen the methodological transparency and presentation quality of the manuscriptt. 
